# Removal of Elemental Mercury from Simulated Flue Gas by a Copper-Based ZSM-5 Molecular Sieve

**Yingbin Zhang [1,†], Jian Zeng [1,†], Liang Xu [1,\*], Xin Liu [2,3], Liangxing Li [1] and Haining Wang [1,2,3,\*]**

1   School of Energy and Mechanical Engineering, Jiangxi University of Science and Technology, Nanchang 330013, China; zhangyingbin@jxust.edu.cn (Y.Z.); 6720190967@mail.jxust.edu.cn (J.Z.); lilx@jxust.edu.cn (L.L.)
2   College of Quality & Safety Engineering, China Jiliang University, Hangzhou 310018, China; 19a0601016@cjlu.edu.cn
3   Jiangxi Key Laboratory of Mining Pollution Controlling, Ganzhou 341000, China
\*   Correspondence: liangxu@hnu.edu.cn (L.X.); 19a0602166@cjlu.edu.cn (H.W.)
†   These authors contributed equally to this work.

**Abstract:** A series of Cu-ZSM-5 molecular sieve adsorbents were prepared by the impregnation method. The experimental results revealed that the mercury removal efficiency of the 2.5 wt% $CuCl_2$-ZSM-5 can reach up to 99%. Furthermore, both the crystal type and pore size distribution of the ZSM-5 molecular sieve remain the same after the process of copper-based active materials impregnated modification, and its specific surface area decreases as the load increases. Importantly, the surface of ZSM-5 modified by $CuCl_2$ equips many more Cu-O functional groups, which are beneficial to the catalytic oxidation of mercury and can even oxidize $Hg^0$ to $Hg^{2+}$. The adsorption process strictly follows the Mars–Maessen reaction mechanism.

**Keywords:** $Hg^0$; ZSM-5 molecular sieve; copper-based; mechanism of mercury removal



## 1. Introduction

Environmental pollution and the energy crisis have always been the focus of our attention [1–5]. The coal industry is the mainstay of the world economy, but the pollutants emitted by it seriously harm the environment. Up until now, the research on the removal of pollutants (such as sulfur dioxide [6], nitrogen oxides [7], and dust [8]) has achieved great success, while there are still few works on removing heavy metals. A major heavy metal pollutant produced by the coal industry, mercury, is as high as 1/3, which emits into the atmosphere and worsens the global environment [9,10]. Therefore, the removal of mercury has become a hot research topic in the field of environmental pollution control.

Recently, a large number of studies have studied mercury removal from coal-fired flue gas. Among them, much interest has been focused on using dedusting or desulfurization and denitrification devices, adsorption, chemical oxidation, and so on [11,12]. Currently, the combination of adsorbent injection and dust removal equipment is considered as one of the most promising prospects for developing removal technology. In addition, many studies on the adsorption of heavy metals by activated carbon have been reported, but it has not been used in engineering, owing to the high cost of mercury removal [13,14]. Consequently, a considerable number of studies have been conducted, leading to the development of an efficient, economical, and environmentally friendly mercury removal adsorbent. We all know that there are three main forms of mercury produced by the coal industry: elemental mercury ($Hg^0$), oxidized mercury ($Hg^{2+}$ or $Hg^+$), and particulate mercury ($Hg^P$) [15]. Both $Hg^{2+}$ and $Hg^P$ can be easily removed by the existing equipment, while $Hg^0$ is difficult to remove from flue gas due to its high volatility, low water solubility, and stable chemical properties [16–18].

Fortunately, studies have shown that elemental mercury can be oxidized by specific oxidizing substances to $Hg^{2+}$, which is easy to be removed [19,20]. Zhang et al. [21] have reported that the Cu-based modification of non-carbon-based adsorbent can improve mercury adsorption and the mercury oxidation rate. Tsai et al. [22] have discovered that the ability of coconut-based activated carbon to adsorb Hg was significantly enhanced after impregnating the coconut-based activated carbon with $CuCl_2$. Du et al. [23] have studied the removal of elemental mercury in flue gas by $CuCl_2$ impregnated neutral $Al_2O_3$, partially artificial zeolite, and activated carbon and found that $CuCl_2$ could provide active Cl for the adsorption reaction of mercury. Moreover, the modified adsorbent has an oxidation efficiency on elemental mercury, and its adsorption capacity has enhanced. All the above studies show that $CuCl_2$ may be a perfect catalyst modification material for mercury adsorption. In addition, the ZSM-5 molecular sieve, as a recyclable, low-cost non-carbon carrier, has been extensively researched in the field of environmental pollution control due to its big specific surface area, perfect adsorption performance, and high thermal stability. [24–26]. Moreover, a previous study shows that ZSM-5 is also a perfect catalyst carrier for mercury adsorption—for instance, the ZSM-5 modified with Mn-Fe mixed oxides can obtain a high mercury removal rate [27]. However, the study of mercury removal by $CuCl_2$-modified ZSM-5 has not been reported. We speculate that the ZSM-5 molecular sieve modified with $CuCl_2$ may have better mercury removal efficiency than both single components.

Therefore, in this study, a variety of copper-based modified ZSM-5 molecular sieve mercury adsorbents were prepared by the impregnation method, and the reaction mechanism of adsorbing $Hg^0$ was deeply explored. The research and development of efficient mercury removal adsorbents has important theoretical significance and engineering value for the removal of mercury from coal-fired flue gas.

## 2. Materials and Methods

The copper-based modified ZSM-5 (Cu-ZSM-5) adsorption materials with different concentrations were prepared by the impregnation of the ZSM-5 molecular sieve and copper-based active materials. We selected $CuCl_2$, $Cu(AC)_2$, $CuSO_4$, and $Cu(NO_3)_2$ as the copper-based active materials. Firstly, the ZSM-5 molecular sieve was pretreated at 600 °C calcination temperature for 2 h. Then, the pretreated ZSM-5 was put into active substance solutions with different concentrations for impregnation for 12 h. After rotary evaporation, it was roasted at 300 °C for 2 h, grinded, and passed through 300 mesh sieves. Finally, the pretreated molecular sieves were put into a vacuum drying box. The ZSM-5 used in this study was purchased from the Catalyst Plant of Nankai University, China, and its ratio of $SiO_2/Al_2O_3$ is 50. The $CuCl_2·2H_2O$, $Cu(AC)_2·H_2O$, $CuSO_4·5H_2O$, and $Cu(NO_3)_2·3H_2O$ were purchased from Tianjin Beichen Founder Reagent Plant, China. All the chemicals used in this article are all of analytical grade.

The mercury removal performances of different copper-ZSM-5 molecular sieves were studied under the same external environment, such as mercury permeability, flow rate, adsorption temperature, and adsorption time. The specific experimental parameters are shown in Table 1. The adsorption experiment was conducted on a small fixed bed adsorption mercury removal system. The experimental system mainly consisted of the mercury vapor generation unit, the catalytic oxidation unit, the mercury concentration analysis unit, and the tail gas treatment unit. The system devices are illustrated in Figure 1.

**Table 1.** Experimental design of the modified ZSM-5 molecular sieve.

| Experimental Materials | wt% | | | | | Experimental Conditions |
|---|---|---|---|---|---|---|
| $CuCl_2$-ZSM-5 | 1.0 | 2.0 | 2.5 | 3.0 | 3.5 | mercury permeability: 500 ng/min |
| $Cu(AC)_2$-ZSM-5 | 0.5 | 1.0 | 1.5 | 2.0 | 2.5 | flow rate: 500 mL/min |
| $CuSO_4$-ZSM-5 | 0.5 | 1.5 | 2.5 | 3.5 | 4.5 | adsorption temperature: 80 °C |
| $Cu(NO_3)_2$-ZSM-5 | 1.0 | 2.0 | 3.0 | 4.0 | 5.0 | adsorption time: 25 min |

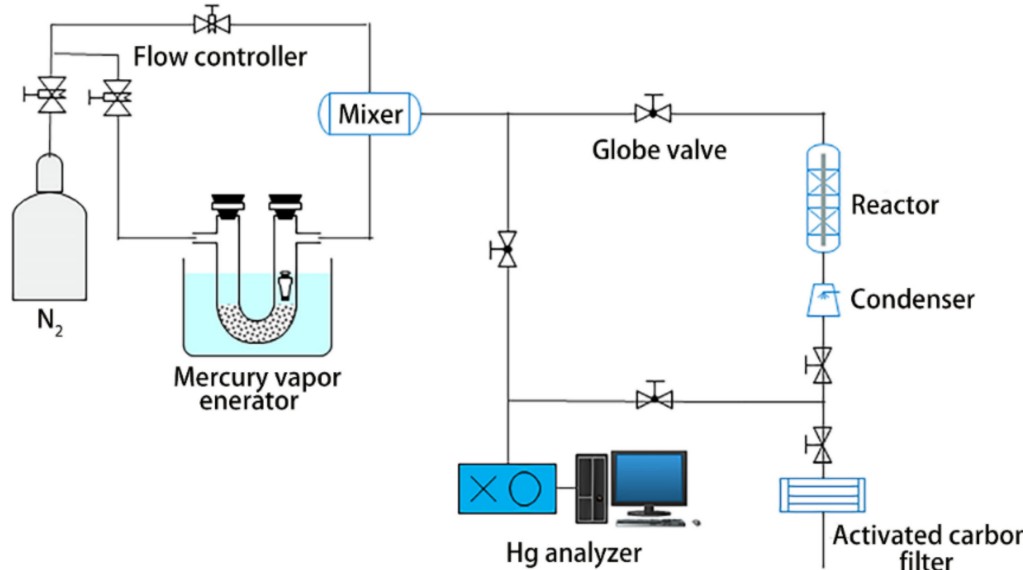

**Figure 1.** Schematic diagram of the mercury adsorption system.

It was assumed that the total mercury generated by the simulated flue gas generation device, the amount of mercury absorbed by the adsorbent through the catalytic oxidation unit, and the amount of mercury absorbed by the analysis unit of the mercury concentration are $q_0$, $q_1$, and $q_2$, respectively. Before the experiment, the system stability was tested to ensure $q_0 = q_1 + q_2$. In the process of the experiment, the remaining mercury was absorbed by the mercury concentration analysis unit after the simulated flue gas passed through the catalytic oxidation unit. The mercury removal performance of the modified ZSM-5 was studied based on the change in mercury content before and after catalytic oxidation, and the mercury removal efficiency ($\eta$) can be evaluated by the following formula:

$$\eta = (q_0 - q_2)/q_0 \times 100\% \tag{1}$$

An X-ray diffractometer (XRD) was utilized to analyze the structural order, cubic structure, and crystallinity of the modified molecular sieves in order to understand the adsorption selectivity and catalytic properties. Electron scanning electron microscopy (SEM-EDS) was performed to analyze the structure tightness of the modified ZSM-5 as well as the types and contents of the constituent elements. An automatic particle size analyzer (BET) was introduced to analyze the particle size and pore size distribution of the modified ZSM-5. X photoelectron spectroscopy (XPS) was used to determine the composition, relative content, and morphology of the adsorbent surface.

### 3. Results

#### 3.1. Mercury Removal Performance

The average mercury adsorption efficiency of the unmodified and modified ZSM-5 under the same standard conditions is shown in Table 2. The results show that the average mercury adsorption efficiency of the unmodified ZSM-5 in 25 min is 42%, and that of the copper-based ZSM-5 is higher than that of the unmodified ZSM-5. Moreover, the mercury removal efficiencies of 2.5% $CuCl_2$-ZSM-5, 1.5% $Cu(AC)_2$-ZSM-5, 2.5% $CuSO_4$-ZSM-5, and 4.0% $Cu(NO_3)_2$-ZSM-5 can even reach 99%, 82%, 79%, and 84%, respectively. Obviously, the loading copper-based active materials in the ZSM-5 molecular sieve can improve the removal efficiency of mercury. The best copper-based modified material is $CuCl_2$, and the optimum load concentration is 2.5 wt%.

**Table 2.** Mercury removal efficiency of different Cu-ZSM-5 molecular sieves.

| Sample | Adsorption Efficiency/η | Sample | Adsorption Efficiency/η |
|---|---|---|---|
| ZSM-5 | 42% | / | / |
| 1.0% CuCl$_2$-ZSM-5 | 94% | 0.5% Cu(AC)$_2$-ZSM-5 | 45% |
| 2.0% CuCl$_2$-ZSM-5 | 97% | 1.0% Cu(AC)$_2$-ZSM-5 | 65% |
| 2.5% CuCl$_2$-ZSM-5 | 99% | 1.5% Cu(AC)$_2$-ZSM-5 | 82% |
| 3.0% CuCl$_2$-ZSM-5 | 98% | 2.0% Cu(AC)$_2$-ZSM-5 | 70% |
| 3.5% CuCl$_2$-ZSM-5 | 97% | 2.5% Cu(AC)$_2$-ZSM-5 | 65% |
| 0.5% CuSO$_4$-ZSM-5 | 48% | 1.0% Cu(NO$_3$)$_2$-ZSM-5 | 66% |
| 1.5% CuSO$_4$-ZSM-5 | 51% | 2.0% Cu(NO$_3$)$_2$-ZSM-5 | 67% |
| 2.5% CuSO$_4$-ZSM-5 | 79% | 3.0% Cu(NO$_3$)$_2$-ZSM-5 | 69% |
| 3.5% CuSO$_4$-ZSM-5 | 67% | 4.0% Cu(NO$_3$)$_2$-ZSM-5 | 84% |
| 4.5% CuSO$_4$-ZSM-5 | 55% | 5.0% Cu(NO$_3$)$_2$-ZSM-5 | 80% |

### 3.2. Characterization of the Cu-ZSM-5 Molecular Sieve

The pore system of the ZSM-5 crystals is formed by the vertical interleaving of the sinusoidal channels and the elliptical straight passages, and its characteristic X-ray diffraction peaks are at 2θ = 23.0°, 24.0°, etc. [28]. The XRD spectra of the ZSM-5 modified by different copper-based materials are shown in Figure 2. It can be seen that the samples prior to and after the modification present the characteristic diffraction peak of the ZSM-5 crystals at 2θ = 23.0°, 24.0°, etc., and significant changes in the characteristic peak are not observed, which indicates that the crystalline form of the ZSM-5 remains unchanged, the framework of the ZSM-5 remains intact, and the copper-based materials are loaded uniformly. However, it should be pointed out that the diffraction peak intensity of the modified ZSM-5 has weakened to some extent compared with the unmodified ZSM-5, illustrating that the copper-based material modified ZSM-5 has an effect on its framework structure. In addition, the analysis of the atlas by Jade software revealed that the crystal diffraction peaks of the copper-modified ZSM-5 at 2θ = 23.0°, 24.0°, etc. deviate from the peaks of the unmodified ZSM-5. The main reason for this may be that the bond length of the Cu-O functional group is longer than that of the Si-O functional group [29], which indicates that Cu has been loaded into the ZSM-5.

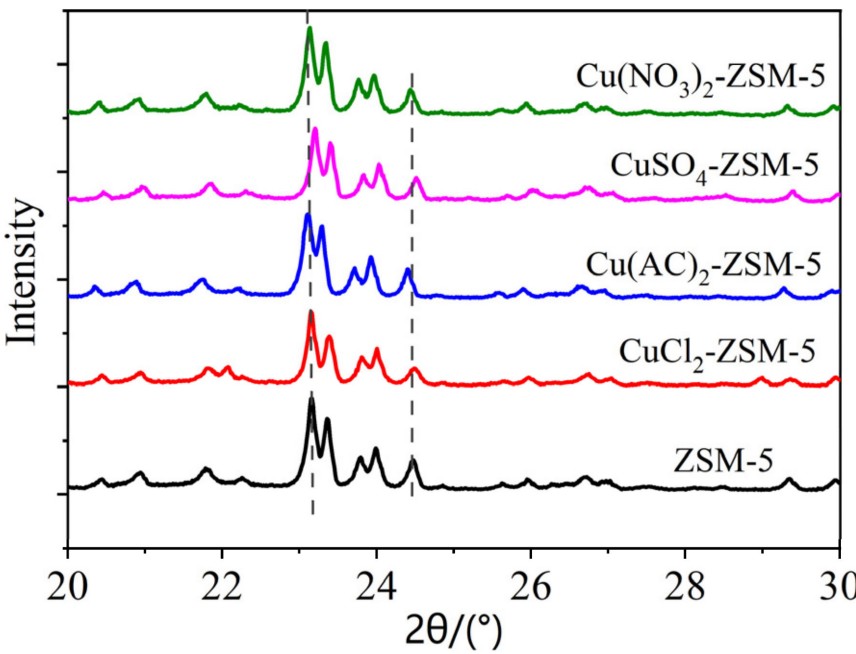

**Figure 2.** Wide angle XRD atlas of ZSM-5 (black), 2.5% CuCl$_2$-ZSM-5 (red), 1.5% Cu(AC)$_2$-ZSM-5 (blue), 2.5% CuSO$_4$-ZSM-5 (pink), and 4.0% Cu(NO$_3$)$_2$-ZSM-5 (green).

Nitrogen adsorption was carried out on molecular sieve samples modified by different concentrations and copper-based materials in order to analyze the pore structure of the adsorbent. Table 3 shows the specific surface area of the ZSM-5 before and after modification, with different mass concentrations and different copper-based materials. It is clearly shown that the specific surface areas of the modified ZSM-5 molecular sieves decrease with the increase in the load concentration of the copper-based materials within a certain range. There are two main reasons for this: (1) some copper-based materials have been loaded to the outer surface of the ZSM-5 molecular sieve, which reduced its outer surface area; (2) parts of the copper-based materials have entered the pores of the molecular sieve and reduced its inner surface area. However, based on the variation trend of the specific surface area of the modified zeolite with different concentrations, when the concentrations of $CuCl_2$-ZSM-5, $Cu(AC)_2$-ZSM-5, $Cu(NO_3)_2$-ZSM-5 are higher than 3.0 wt%, 2.0 wt%, and 4.0 wt%, respectively, their specific surface area did not decrease but rather rose. This indicates that the load of Cu in the ZSM-5 has saturated. A large number of copper-based materials accumulate on the outer surface of the adsorbent, resulting in surface agglomeration and condensation, which increases the surface area.

**Table 3.** Specific surface area of different Cu-ZSM-5 molecular sieves.

| Sample | Specific Surface Area ($m^2 \cdot g^{-1}$) | Sample | Specific Surface Area ($m^2 \cdot g^{-1}$) |
|---|---|---|---|
| ZSM-5 | 295.7195 | / | / |
| 1.0% $CuCl_2$-ZSM-5 | 265.0384 | 0.5% $Cu(AC)_2$-ZSM-5 | 264.7255 |
| 2.0% $CuCl_2$-ZSM-5 | 232.3853 | 1.0% $Cu(AC)_2$-ZSM-5 | 246.8681 |
| 2.5% $CuCl_2$-ZSM-5 | 197.3325 | 1.5% $Cu(AC)_2$-ZSM-5 | 233.8920 |
| 3.0% $CuCl_2$-ZSM-5 | 177.0272 | 2.0% $Cu(AC)_2$-ZSM-5 | 168.9154 |
| 3.5% $CuCl_2$-ZSM-5 | 182.3033 | 2.5% $Cu(AC)_2$-ZSM-5 | 215.9531 |
| 0.5% $CuSO_4$-ZSM-5 | 279.6190 | 1.0% $Cu(NO_3)_2$-ZSM-5 | 266.8362 |
| 1.5% $CuSO_4$-ZSM-5 | 258.7413 | 2.0% $Cu(NO_3)_2$-ZSM-5 | 238.6340 |
| 2.5% $CuSO_4$-ZSM-5 | 253.4128 | 3.0% $Cu(NO_3)_2$-ZSM-5 | 236.8589 |
| 3.5% $CuSO_4$-ZSM-5 | 240.4521 | 4.0% $Cu(NO_3)_2$-ZSM-5 | 201.5803 |
| 4.5% $CuSO_4$-ZSM-5 | 222.4817 | 5.0% $Cu(NO_3)_2$-ZSM-5 | 228.0537 |

According to the above analysis of the specific surface areas, it can be seen that, as the concentration of copper-based active material increases, the amount of copper loaded on the ZSM-5 also increases, thereby promoting the mercury removal efficiency. However, when the copper loading reaches a certain amount, due to the effects of surface agglomeration and condensation, the mercury removal efficiency no longer increases but rather decreases. In addition, in combination with Tables 2 and 3, the four types of copper-based active material modified ZSM-5 all achieve the best mercury removal efficiency before the copper load is saturated, which further shows that excessive copper loading will reduce the mercury removal efficiency.

Figure 3a shows the nitrogen adsorption and desorption curves of ZSM-5, 2.5% $CuCl_2$-ZSM-5, 1.5% $Cu(AC)_2$-ZSM-5, 2.5% $CuSO_4$-ZSM-5, and 4.0% $Cu(NO_3)_2$-ZSM-5. It can be seen from the diagram that the adsorption isotherm is similar to the I type (Langmuir) adsorption isotherm at a relatively low pressure, and the adsorption process belongs to the single molecular layer adsorption, which indicates the existence of microporous adsorption. With the increase in relative pressure, the adsorption process changes from monolayer to multilayer, which indicates that there are some mesopores in the adsorbent. When the relative pressure is 0.5–1.0, the adsorption isotherm belongs to the typical IV-type adsorption isotherm containing mesopores. The molecular sieves before and after modification have the hysteresis loop related to the mesopore, but the hysteresis loop is not obvious, which shows that there are only a few mesopores in the molecular sieve.

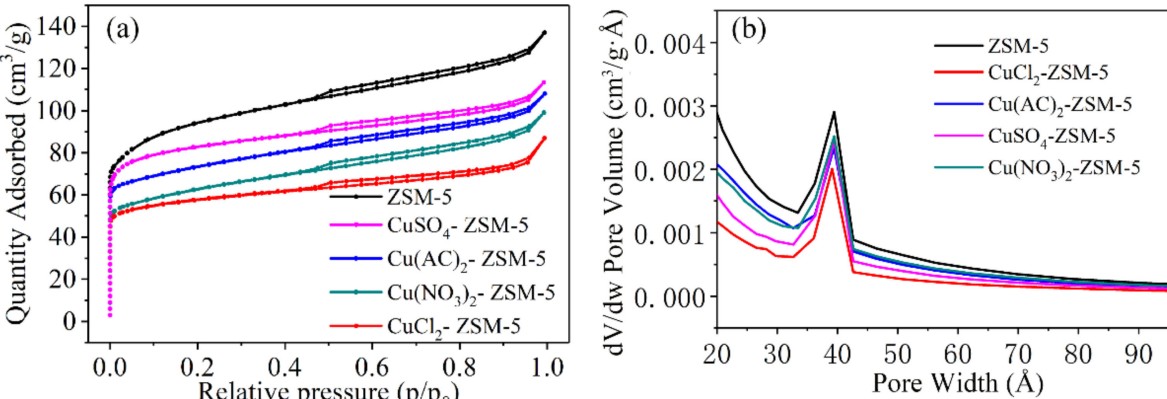

**Figure 3.** (**a**) Nitrogen adsorption and desorption curves of ZSM-5, 2.5% CuCl$_2$-ZSM-5, 1.5% Cu(AC)$_2$-ZSM-5, 2.5% CuSO$_4$-ZSM-5, and 4.0% Cu(NO$_3$)$_2$-ZSM-5, as well as their (**b**) aperture distribution curves.

Internationally, the apertures are generally divided into three types: micropore (the aperture is less than 2 nm), mesopore (the aperture is between 2 and 50 nm), and macropore (the aperture is more than 50 nm) [30,31]. The pore size distribution of the ZSM-5 before and after copper modification is shown in Figure 3b. We can find that the peak position of the ZSM-5 does not change after the modification by copper-based active materials, indicating that the load materials basically do not block the channel of the ZSM-5 molecular sieve and have little influence on the distribution of the pore size. However, the intensity and width of the peak have been changed, which indicates that the range of the aperture has been changed. The higher the intensity of the peak is, the higher the number of the aperture is; and the narrower the width of the peak is, the more uniform the size of the aperture is. Clearly, both the modified and unmodified ZSM-5 molecular sieves have micropores, mesopore, and macropore distributions. The mesoporous peak intensity of the copper-based material modified ZSM-5 is less than that of the unmodified ZSM-5, and the mesoporous peak of CuCl$_2$-ZSM-5 is the lowest and narrowest, indicating that the CuCl$_2$-ZSM-5 molecular sieve has fewer mesopores and that uniform particle size distribution is more conducive to mercury removal.

The SEM and surface energy spectra of the modified ZSM-5 after mercury adsorption are shown in Figure 4. Obviously, the Cu element has been successfully loaded onto the ZSM-5, and the mercury has been adsorbed in the molecular sieve. The peak intensity of the electron spectra of Cu and Hg was quite different, which indicates that the contents of Cu and mercury in the ZSM-5 are different. We can compare the modification effects of different copper-based active materials by measuring the relative content of Cu and adsorbed mercury. As shown in Table 4, the abilities of the CuCl$_2$-ZSM-5, Cu(AC)$_2$-ZSM-5, CuSO$_4$-ZSM-5, and Cu (NO$_3$)$_2$-ZSM-5 molecular sieves for Hg adsorption are 5.9%, 1.4%, 3.9%, and 3.6%, respectively. There is no doubt that the CuCl$_2$-ZSM-5 equips a relatively high adsorption property of mercury.

In order to obtain the detailed information about Cu and Hg after the adsorption of the modified ZSM-5 molecular sieve and analyze the mechanism of mercury adsorption further, the Cu and Hg elements in the CuCl$_2$-ZSM-5 were quantitatively analyzed by XPS [32]. Figure 5 is the XPS full spectrum of the CuCl$_2$-ZSM-5 molecular sieve after mercury adsorption. The graph shows that the peak of the Cu2p spectrum appeared between the binding energy of 930–960 eV, and the peak of the Hg4f spectrum appeared between 93–112 eV. The above analysis further shows that Cu has successfully loaded onto the ZSM-5 after modification, and Hg has also adsorbed in the adsorbent after the adsorption experiment. In addition, we performed selected-area EDX analysis on the chemical composition of CuCl$_2$-ZSM-5 combined with energy dispersive X-ray spectroscopy to confirm the uniform distribution of copper. As shown in Figure 6, the contrasting Cu species are dispersed on

the entire surface of ZSM-5, which indicates that Cu is uniformly dispersed on the surface of the ZSM-5 zeolite.

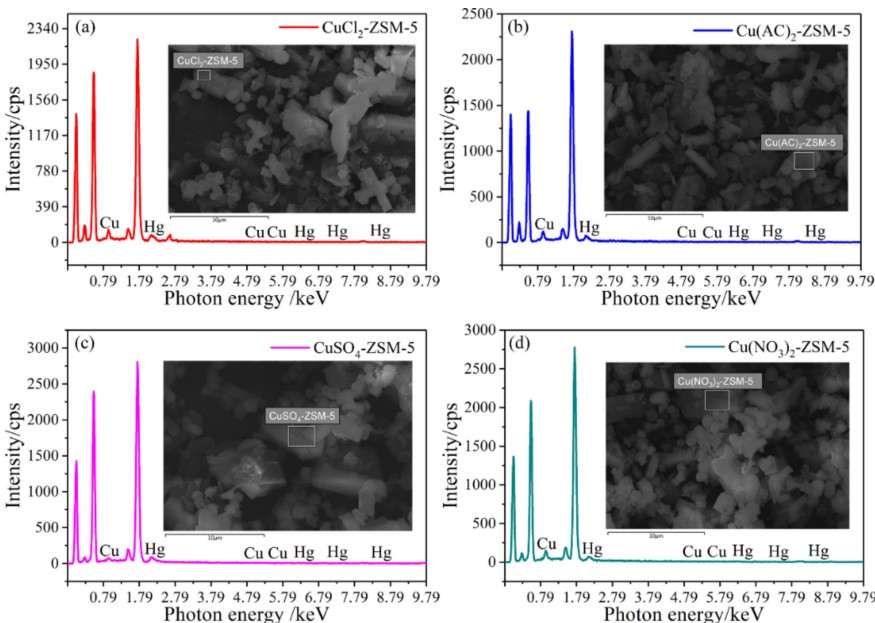

**Figure 4.** SEM and surface energy spectra of (**a**) 2.5% $CuCl_2$-ZSM-5, (**b**) 1.5% $Cu(AC)_2$-ZSM-5, (**c**) 2.5% $CuSO_4$-ZSM-5, and (**d**) 4.0% $Cu(NO_3)_2$-ZSM-5.

**Table 4.** Element type and content of 2.5% $CuCl_2$-ZSM-5, 1.5% $Cu(AC)_2$-ZSM-5, 2.5% $CuSO_4$-ZSM-5, and 4.0% $Cu(NO_3)_2$-ZSM-5.

| Sample | Atomic Percentage (At%) | |
| --- | --- | --- |
| | **Cu** | **Hg** |
| $CuCl_2$-ZSM-5 | 94.1 | 5.9 |
| $Cu(AC)_2$-ZSM-5 | 98.6 | 1.4 |
| $CuSO_4$-ZSM-5 | 96.1 | 3.9 |
| $Cu(NO_3)_2$-ZSM-5 | 96.4 | 3.6 |

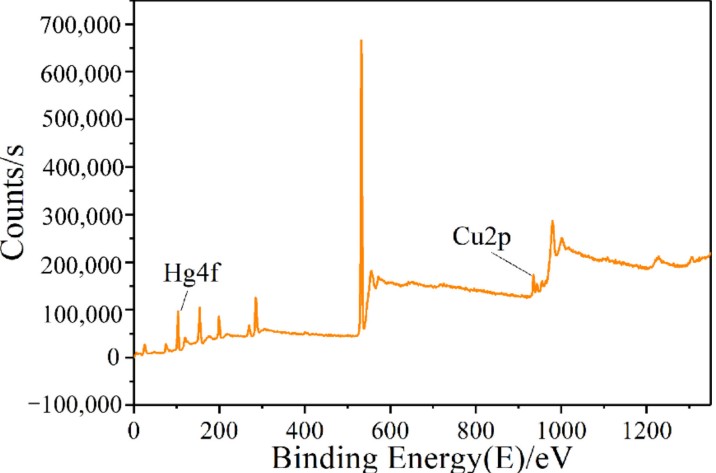

**Figure 5.** XPS full spectrum of 2.5% $CuCl_2$-ZSM-5.

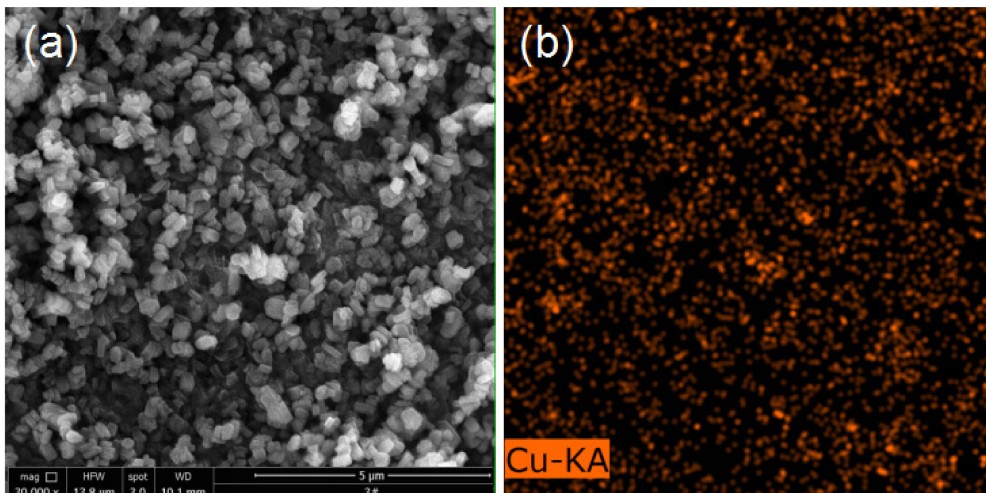

**Figure 6.** (**a**) SEM image of 2.5% CuCl$_2$-ZSM-5, and (**b**) the corresponding EDX elemental mapping images of Cu.

Figure 7a was the XPS fine spectrum of Cu2p. According to the standard track peak, the binding energies of Cu2p1/2, Cu2p3/2, sat, and Cu2p3/2 are between 952.1–954.5 eV, 941.9.1–944.8 eV, and 932.1–936.5 eV, respectively. According to the XPS-peak-differentiating analysis, two peaks of 952.98 eV and 955.18 eV appear at the high binding energy Cu2p1/2, which is the peak of the low-price Cu, indicating that there are two valence states of Cu$^0$ and Cu$^+$ in the surface of the adsorbents. A peak of 942.38 eV appears at the peak of Cu2p3/2, sat, which is the characteristic satellite peak corresponding to Cu$^{2+}$, and two peaks of 933.18 eV and 935.38 eV, corresponding to the characteristic peaks of Cu$^+$ and Cu$^{2+}$, appear at the broad and strong main peaks of the Cu2p3/2 photoelectron. In addition, Figure 7b shows the XPS fine spectrum of Hg4f. Clearly, there are two peaks of Hg$^{2+}$ that appear at 102.23 eV and 103.13 eV, while the Hg$^0$ energy spectrum is not detected at 99.9 eV. Therefore, this strongly proves that the CuCl$_2$-ZSM-5 molecular sieve has the oxidizing properties for Hg$^0$, oxidizing the adsorbed Hg$^0$ to Hg$^{2+}$, and the Hg element is adsorbed in the CuCl$_2$-ZSM-5 molecular sieve in an oxidized form.

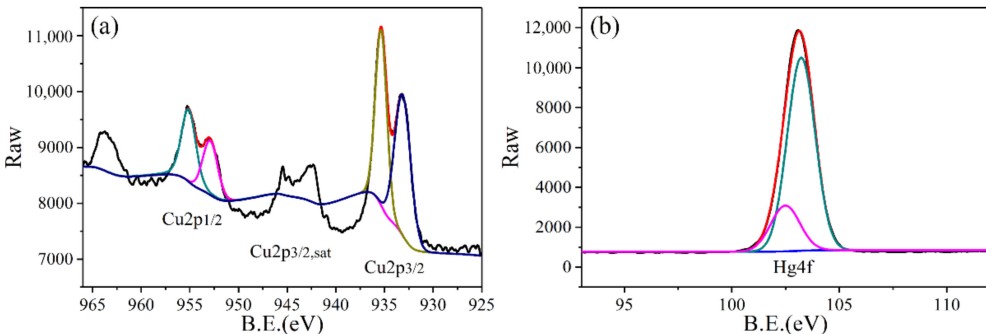

**Figure 7.** XPS fine spectrum for (**a**) Cu2p and (**b**) Hg4f of 2.5% CuCl$_2$-ZSM-5.

### 3.3. Mechanism of Mercury Removal by the CuCl$_2$-ZSM-5 Molecular Sieve

Based on the experiment and the characterization results, the reaction mechanism of Hg$^0$ adsorption in CuCl$_2$-ZSM-5 molecular sieves can be speculated upon. In the adsorption reaction process at the gas–solid interface, both physical adsorption and chemisorption have occurred. Firstly, the reaction process of gaseous Hg$^0$ during the adsorption process is as follows:

$$Hg^0(g) \rightarrow Hg^0(ad) \tag{2}$$

1. Part of the $Hg^0$ (ads) adsorbed by the Cu-ZSM-5 molecular sieve will directly undergo a redox reaction with the loaded oxidizing material $CuCl_2 \cdot 2H_2O$. The reaction formula is as follows:

$$Hg^0(ad) + CuCl_2 \cdot 2H_2O \rightarrow HgCl_2(ad) + Cu(ad) + 2H_2O \tag{3}$$

2. At the same time, the other part of $Hg^0$ (ads) adsorbed by the Cu-ZSM-5 molecular sieve will indirectly undergo a redox reaction with the supported oxidizing material $CuCl_2 \cdot 2H_2O$. The reaction formula is as follows:

$$CuCl_2 \cdot 2H_2O + 1/2O_2 \rightarrow CuO + 2Cl^* + 2H_2O \tag{4}$$

$$3CuCl_2 \cdot 2H_2O + O_2 \rightarrow CuO + Cu_2O + 6Cl^* + 6H_2O \tag{5}$$

$$Hg(ad) + Cl^* \rightarrow HgCl(ad) \tag{6}$$

$$HgCl(ad) + Cl^* \rightarrow HgCl_2(ad) \tag{7}$$

$$2CuO + Hg \leftrightarrow Cu_2O + HgO \tag{8}$$

The above reactions indicate that, after loading the $CuCl_2$ active substance in the ZSM-5 molecular sieve, $Hg^0$ has a heterogeneous catalytic oxidation reaction, which is in accordance with the Mars–Maessen reaction mechanism proposed by Granite E. J. et al. [33]. Corma A. et al. [34] pointed out that the adsorption capacity of the molecular sieve to the adsorbate basically depends on the size of the adsorbate. Among the four materials, the relative molecular mass of $CuCl_2$ is the smallest. Therefore, under the same conditions, the ZSM-5 molecular sieve can load more $CuCl_2$ active components so as to a achieve better modification effect. The $Hg^0$ is adsorbed by the adsorbent, which reacted with the oxidant in the lattice. This reaction mechanism could account for the adsorption mechanism of $Hg^0$ in $CuCl_2$-ZSM-5. Therefore, the modification of ZSM-5 with $CuCl_2$ can enhance the oxidation performance of ZSM-5 and improve the removal efficiency of $Hg^0$.

## 4. Conclusions

This work reports a simple impregnation method to prepare copper-based modified ZSM-5 adsorbents with different concentrations. The results show that copper-based active materials, such as $CuCl_2$, $Cu(AC)_2$, $CuSO_4$, and $Cu(NO_3)_2$, can modify the ZSM-5 molecular sieve and improve its mercury removal performance, and the $CuCl_2$-ZSM-5 molecular sieve has a better adsorption performance of $Hg^0$—especially the 2.5 wt% $CuCl_2$ modified ZSM-5 molecular sieve, which can even achieve 99% mercury removal efficiency. Moreover, we found that, after loading the active material, the specific surface area of the ZSM-5 was reduced, and the number of pores changed, but the pore size distribution hardly changed. After the adsorption of $Hg^0$, the skeleton of the ZSM-5 can remain intact. In addition, by analyzing the adsorption mechanism of $Hg^0$ in a Cu-ZSM-5 molecular sieve, $Hg^0$ can be oxidized to $Hg^{2+}$ and easily removed, which belongs to the Mars–Maessen reaction mechanism. This research can provide new insights for the removal of mercury in coal flue gas and promote a clean environment.

**Author Contributions:** Conceptualization, L.X. and H.W.; methodology, Y.Z. and J.Z.; validation, Y.Z. and J.Z.; formal analysis, X.L.; investigation, L.X.; resources, L.X. and H.W.; data curation, Y.Z. and J.Z.; writing—original draft preparation, Y.Z.; writing—review and editing, Y.Z. and J.Z.; visualization, Y.Z.; supervision, L.X. and L.L.; project administration, L.X., L.L. and H.W.; funding acquisition, Y.Z., L.X. and H.W. All authors have read and agreed to the published version of the manuscript.

**Funding:** This study is supported by the Scientific Research Fund of the Jiangxi Provincial Education Department (No. GJJ180493), the Jiangxi Provincial Natural Science Foundation (No. 20212BAB201013, 2019BAB206019), and the Natural Science Foundation of China (No. 51568024).

**Institutional Review Board Statement:** Not applicable.

**Informed Consent Statement:** Not applicable.

**Data Availability Statement:** The data are available in a publicly accessible repository.

**Conflicts of Interest:** The authors declare no conflict of interest.

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
