# Peer review of "Removal of Elemental Mercury from Simulated Flue Gas by a Copper-Based ZSM-5 Molecular Sieve"

_coatings, doi:10.3390/coatings12060772_

Round 1

Reviewer 1 Report

In this study, the authors prepared ZSM-5 sieve modified with CuCl2 for potential elemental Hg capture. In general, the article can be considered for publication after a major revision. 

1) The text should be polished by a native speaker. The preparation methodology is not reproducible, supply synthesis details and reagents purity. 
2) Provide TEM images instead of SEM, this will confirm the porous structure formation. 
3) EDX elemental mapping over a selected area should be shown for selected samples to confirm the uniform distribution of the Cu. 
4) EDX is a semiquantitative method, hence, Cu and Hg concentrations should be tested by more quantitative methods such as ICP-MS/OES or AAS. 
5) Reusability of molecular sieve and long-time operation stability should be tested.  

Reviewer 2 Report

Journal: Coatings

Title: Removal of Elemental Mercury from Simulated Flue Gas by ZSM-5 Modified with CuCl2

The manuscript is interesting, it deals with the impregnation of ZSM-5 zeolite with various copper salts as a means for impregnation of zeolite. Modified ZSM-5 samples were then used for removal of gaseous elemental mercury. The authors proposed a reaction mechanism for the removal of mercury, concluding that Cu has a catalytic property, the oxidation of Hg(0) to Hg2+, whereby Hg2+ is more easily adsorbed on modified zeolite.

The manuscript deserved to be published after appropriate corrections.

Comments follow:

The title of the manuscript is not the most appropriate as the Authors did not perform the modification exclusively with CuCl2.

There are a lot of typos in the manuscript, for example words separated by dashes (-), lines 17, 18, 25, 30, 48, 54, 99, 140, 158, 213

It is necessary in correct way write chemical formulas and symbols (superscript, subscript), lines: 113, 116, 187, 188, 220, 221, 230, 231, 232, 237 (Hg0, CuCl2-ZSM-5), 241, 253

Table 1, Table 2, Figure 3, Figure 4, Figure 5, Figure 6 are not in the appropriate place. They appear either in the wrong section or in the wrong place, before the text. It is necessary to first announce in the text Table or Figure and then they appear.

Figure 2, Figure 3, Figure 4, Table 4, Figure 5, Figure 6 do not have an adequate name. It is necessary to emphasize which material they refer to.

Line 79-81: „The mercury removal performance of different copper-ZSM-5 molecular sieves were studied under the same external environment, such as mercury permeability, flow rate, adsorption temperature and adsorption time.“ - It is necessary to specify the performance of these 4 experiments influencing mercury permeability, flow rate, adsorption temperature and adsorption time. Furthermore, from the result of line 11: "in 25 min is 42%." it is observed that the results in Table 2 refer only to the influence of contact time.

It is necessary to give the results of the other 3 experiments.

The authors found that CuCl2 modifier is the most effective, but there is no explanation why it is the most effective. What is the effect of different anions (Cl-, SO42-, AC, NO3-) on Cu sorption on ZSM-5?

Lines 155-157, 160-161, 260-261: Define the reason why this is so.

In Figure 3, on the abscissa and the ordinate write the symbols for the units of measurement in the same way.

It would be useful to attach the results of the SEM-EDS and XPS of starting and unsaturated samples with Hg, especially to see insight into the distribution of Cu on surface and whether its leaching from the ZSM-5 surface occurs during the sorption experiment.

Line 238: „CuCl 2 ·2H2O“ - In what form is Cu on the surface of ZSM-5 ??

Have you taken reactions (3) - (8) from the literature? Reactions (4), (5) and (8) are not balanced.

In the end, it would be good to look at what to do with saturated zeolite with Hg. Disposal, regeneration ....

Reviewer 3 Report

The subject of the proposed paper is environmental pollution control. The list of emerging pollutants has increased as a direct consequence of anthropogenic activities in the last decades, but heavy metals are still consistently high on the list. So, the removal of heavy metals such as mercury, lead or cadmium from the environment is a field of high importance. The article is well written, based on recent literature and the results presented in this paper are well prepared. However, there are still some significant aspects that authors should improve. There are a lot of studies about the removal of mercury, including the application of zeolites for this purpose. Why Authors decided to choose the specific ZSM-5, having multiple choices of adsorbents? The motivation to use the copper-based adsorbents is sufficient, but it seems the discussion has been missed? The Authors should emphasize the advantages and disadvantages of this kind of modification in comparison to recent papers. The aim of the work should be highlighted, and it seems that it should be a bit more justified.

  • The introduction section looks like a discussion section (which could be used for this purpose). 
  • Table 1 is confusing. The experimental conditions could be below the table as a description.
  • The information on the chemical used is scant, the detailed information should be provided.
  • There are many typographical and logical errors or inaccuracies. The English should be carefully checked.

In my opinion, the manuscript requires major revision before any other consideration.

Round 2

Reviewer 1 Report

A revised manuscript can be accepted for publication

Reviewer 2 Report

The manuscript has been upgraded, it can be accepted after the mentioned corrections.

Table 2 must be within section 3.1.

Equation 5 is again unbalanced, Cu moles are not balanced!

Reviewer 3 Report

I would like to thank the Authors for considering my suggestions.